# Improving Length-Generalization in Transformers via Task Hinting

**Pranjal Awasthi**
Google Research
pranjalawasthi@google.com

**Anupam Gupta**
Carnegie Mellon University and Google Research
anupamg@cs.cmu.edu

## Abstract

It has been observed in recent years that transformers have problems with *length generalization* for certain types of reasoning and arithmetic tasks. In particular, the performance of a transformer model trained on tasks (say addition) up to a certain length (e.g., 5 digit numbers) drops sharply when applied to longer instances of the same problem. This work proposes an approach based on *task hinting* towards addressing length generalization. Our key idea is that while training the model on task-specific data, it is helpful to simultaneously train the model to solve a simpler but related auxiliary task as well.

We study the classical *sorting* problem as a canonical example to evaluate our approach. We design a multitask training framework and show that models trained via task hinting significantly improve length generalization. In particular, for sorting we show that it is possible to train models on data consisting of sequences having length at most 20, and improve the test accuracy on sequences of length 100 from less than 1% (for standard training) to more than 92% (via task hinting).

Our study uncovers several interesting aspects of length generalization. We observe that while several auxiliary tasks may seem natural *a priori*, their effectiveness in improving length generalization differs dramatically. We further use probing and visualization-based techniques to understand the internal mechanisms via which the model performs the task, and propose a theoretical construction consistent with the observed learning behaviors of the model. Based on our construction, we show that introducing a small number of length dependent parameters into the training procedure can further boost the performance on unseen lengths. Finally, we also show the efficacy of our task hinting based approach beyond sorting, giving hope that these techniques will be applicable in broader contexts.

## 1 Introduction

Large transformer models trained on massive datasets continue to demonstrate impressive capabilities across a range of tasks in language understanding, image modeling and other domains (Radford et al., 2019; Brown et al., 2020; Chowdhery et al., 2022; Chen et al., 2022; Tu et al., 2023). At the same time there is a growing body of work on the limitations and vulnerabilities of such models. This work concerns the *length generalization* problem. For many natural tasks—especially ones involving multi-step reasoning such as addition, multiplication, program execution etc.—there is a natural notion of the *length* of an input, e.g., the number of digits when performing the addition task (Anil et al., 2022). It has been observed that the performance of transformers on such tasks drops sharply when tested on instances with lengths not seen during training (Nye et al., 2021; Zhang et al., 2022; Jelassi et al., 2023; Abbe et al., 2023). As formalized in Abbe et al. (2023) this phenomenon can also be studied as an extreme form of out-of-distribution (OOD) robustness where the support of the test-set distribution is disjoint from that of the training distribution.

Current approaches to tackle length generalization can be broadly divided into two categories. One set of recent works start with a pre-trained large language model (LLM) and investigate fine-tuning/in-context learning for extrapolating to larger lengths. Most notably, the works of Wei et al. (2022); Anil et al. (2022) observe that in-context learning strategies such as *chain-of-thought prompting* and *scratchpad prompting* can help improve the out-of-distribution performance

of LLMs. Another set of works consider case studies on simpler tasks, and perform task-specific training to improve length generalization (Zhang et al., 2022; Jelassi et al., 2023; Abbe et al., 2023).

Our work falls into the second category: our goal is to develop general-purpose training techniques to improve length generalization. To put the challenges underlying length-generalization in context, let us list several natural approaches that do not seem to help. For instance, it was observed in Anil et al. (2022) that simply scaling the model and data-set sizes alone does not suffice for length generalization. The authors also observed that while scratchpad prompting helps during in-context learning, fine-tuning a pre-trained LLM on scratchpads does not seem to work, which is surprising. Similarly, the authors in Zhang et al. (2022) observed that while using a pre-trained BERT (Devlin et al., 2018) model helps improve the performance on the LEGO task that the authors introduced in the work, the improvements are limited and not enough to address the problem beyond a certain point. Hence training-time techniques to address the length generalization problem either introduce task-specific architectures (Zhang et al., 2022) or perform *data priming/curriculum learning*, where data from higher-length instances is introduced into the training procedure (Jelassi et al., 2023; Abbe et al., 2023). Interestingly, the authors in Jelassi et al. (2023) observed that while introducing a small amount of training data from instances of length $n$ may help in generalizing to test instances also of length $n$, the model could fail completely on test instances of length $n + 1$!

Motivated by the above works, we study whether there are general-purpose training techniques for improving length generalization. As in prior works, we focus on some simple tasks as our use-cases, and consider training transformer models from scratch. For most of the paper we consider the classical problem of *sorting* as a canonical example. Given an unsorted sequence of natural numbers of length $n$, we consider training decoder-only transformer models to learn to output the sorted sequence. We work with standard transformer architectures and explicitly refrain from using even a small amount of data from higher length sequences, either during training or for model selection. Our main contribution is the framework of *task hinting* for tackling length generalization. Our approach rests on the core idea that as humans, learning to solve a particular task also involves learning to solve simpler useful sub-tasks. For instance, a human student who claims to sort numbers well is also expected to know how to compare two numbers, identify successor/predecessor of a number in a sequence, count the number of occurrences of a number in a sequence and so on.

Hence we propose a multi-task learning framework where the transformer network is trained simultaneously to solve a *main* task (such as sorting) and an *auxiliary task* (such as identifying the successor element). We show that this approach leads to a powerful framework for improving length generalization. In particular, we demonstrate that by training the model only on data of up to length $n = 20$ sequences, one can improve the test accuracy on length $n = 100$ sequence from less than $1\%$ (for standard training) to more than $92\%$ (via task hinting). In the second part of the paper we perform a deeper investigation of when and how task hinting helps. We observe that while many auxiliary tasks may seem natural *a priori*, their effect on length generalization varies greatly. For the task of sorting sequences, we observe that the task of identifying the successor element helps the most, while the task of counting helps the least.

We further use visualization techniques to conclude that for each task the transformer network tends to be biased towards a particular mechanism for solving the task. Perhaps naturally, auxiliary tasks that align well with this bias tend to help the most. We identify certain computational primitives that the network tends to implicitly capture at various layers and propose a theoretical construction of a sorting transformer that is consistent with the empirical findings. Based on our theory we identify a small number of *length-dependent* parameters whose introduction into the model boosts the length generalization of transformers significantly (even for models that are trained without task hinting). Finally, we demonstrate the effectiveness of our proposed framework for another simple task namely that of incrementing a number. The results for this task can be found in Appendix C.

## 2 RELATED WORK

Length generalization in transformers is a challenging problem, with several confounding factors, such as the role of positional embeddings, architectural choices, and dataset formatting and/or prompting strategies. The works of Dubois et al. (2019); Press et al. (2021) propose modifications to the standard attention mechanism to enable length extrapolation. The work of Newman et al.

```
Input:   5  3  1  4  2  ⊥  1  2  3  4  5  PAD  PAD  ....
Mask:    0  0  0  0  0  1  1  1  1  1  0   0    0   ....
```

Figure 3.1: *An example input sequence for decoder only model training. The mask ensures that we only penalize the model for predictions at the output positions.*

(2020) observes a surprising role played by the presence/absence of the EOS token. In particular, they observe the models without the EOS token extrapolate significantly better to higher lengths.

The work of Anil et al. (2022) explores in-context learning strategies for improving length generalization. The authors show that length generalization can be significantly improved for tasks such as parity and variable assignment by prompting via scratchpads. They also observe certain counterintuitive behaviors, such as the lack of improvements in length generalization when fine-tuning a model via scratchpad prompts. While it is conceivable that length generalization can be improved via more complex scratchpads/chain-of-thoughts, augmenting a training dataset with such prompts may not always be feasible, and may lead to a significant blow-up of the input context length (Malach, 2023). As another example, the recent work of Liu & Low (2023) fine-tunes an open source LLaMA model (Touvron et al., 2023) for multi-digit multiplication via scratchpad/chain-of-thought training. It observes that while in-distribution accuracy significantly improves, the resulting models continue to suffer from length generalization.

The work of Zhang et al. (2022) proposes a LEGO task that has a similar flavor to the task of sorting. The authors observe that when training a BERT model from scratch for length $n = 6$, the in-distribution accuracy is $100\%$, but the accuracy for $n = 8 \ldots 12$ is no better than random. Moreover, they show that training a specific architecture, namely the ALBERT model (Lan et al., 2019), improves the length generalization to some extent. In Jelassi et al. (2023) the authors propose the idea of *data priming* for length generalization. This involves introducing a small amount (less than $1\%$) of the data from higher lengths (i.e., the test distribution) into the training process to improve the out of distribution performance. However, the authors observe that priming a dataset for an unseen length $n$ may not have any benefits for performance at length $n + 1$. In a similar vein, the authors in Abbe et al. (2023) propose a *curriculum learning* procedure, where data from higher and higher lengths are gradually improved into the training procedure.

Our work also involves understanding the internal learning mechanisms of the trained models via simple projection based techniques. In a similar vein, the recent work of Nanda et al. (2023a) studies a depth-one network trained for addition modulo 113, using $d = 128$-dimensional representations. Using the structured nature (and limited size) of the task, they show how zooming in on neurons and analyzing network weights can help understand the underlying mechanisms. Another set of recent works (e.g., by Li et al. (2022); Nanda et al. (2023b)) use *probing* to find mappings from the internal representations of networks to the actual external state of the problem (Othello) being solved. The focus of our work is on showing that broad-spectrum techniques—based on simple projections onto the embedding and unembedding bases—can result in surprisingly valuable insights.

Finally, our work uses the framework of multitask learning that has a rich literature (Crawshaw, 2020). Traditionally, multitask learning is used for obtaining good representations that can adapt to new auxiliary tasks using small amounts of additional data. In contrast, in this work we use multitask learning primarily to improve the out-of-distribution robustness of the main task itself.

## 3 SORTING

For the majority of the paper we focus on sorting as our canonical example. We consider solving this task via decoder-only transformer models (Brown et al., 2020) trained from scratch. We work with a vocabulary $\Sigma$ of integers from 1 to 100 and introduce two additional tokens, $\perp$ as the end-of-input delimiter, and PAD as a padding token to ensure that all input sequences during training have the same length. Given an input sequence, we train a decoder-only causal transformer model to predict the sorted sequence one token at a time. The training is done via the standard next-token prediction framework with the cross-entropy loss. See Figure 3.1 for an example input sequence, and the mask we use to penalize the model only for the output positions.

Our training dataset consists of sequences of lengths up to 20, where each sequence is formed by drawing numbers from $\Sigma = \{1, 2, \ldots, 100\}$ uniformly at random with replacement. Furthermore, to simulate the realistic setting where data freely available on the internet is biased towards shorter sequence lengths, we ensure that $80\%$ of the training set consists of sequence lengths from $\{2, 3, 4, 5\}$, and the remaining $20\%$ consists of sequence lengths $\{6, 7, \ldots, 20\}$. We first investigate whether (and by how much) scaling the data and model can help with length-generalization. To do this, we train a depth-2 model (see Appendix D for the hyperparameter settings) with dataset sizes of $\{1M, 10M, 40M, 160M\}$, and we also train models with depths $\{2, 4, 8, 12\}$ on a $1M$ training set. All the models are trained using the Adam optimizer (Kingma & Ba, 2014) for $100k$ gradient steps[1] with a batch size of 1024, and a one-cycle cosine learning rate (Loshchilov & Hutter, 2016) starting with the base learning rate of $1e - 5$. (We use the first 10 epochs for a linear warmup to the base learning rate.) When evaluating the models, we use greedy decoding.

The result of data scaling is shown in Figure 3.2. While more data helps to some extent—the test accuracy on length 50 sequences improves from $64\%$ to close to $90\%$—further scaling does not help and all the models achieves less than $1\%$ accuracy on length 100 sequences. Here test accuracy refers to the fraction of the sequences in the test set ($100k$ examples per sequence length) where the model outputs the correct sorted sequence. Similarly, scaling the depth from 2 to 4 helps improve the accuracy on length 50 sequences, but we do not observe any further benefits (Figure 3.3) thereafter. Again, the accuracy for length 100 sequences is less than $1\%$ for all model and data sizes. This is consistent with the behavior observed in Nye et al. (2021): model and data scaling alone does not seem enough to tackle length-generalization.

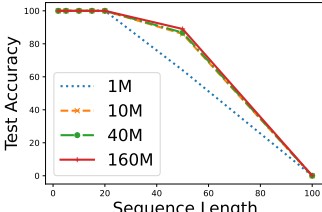

Figure 3.2: Effect of data scaling on length generalization. While performance improves on length 50 sequences, there is no benefit at higher lengths.

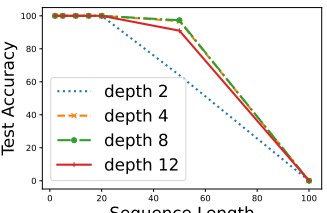

Figure 3.3: Effect of model scaling on length generalization. All the models have less than $1\%$ test accuracy for length 100 sequences.

## 3.1 TASK HINTING

We now introduce the framework of *task hinting*. We consider a multi-task setup where we train the model to simultaneously perform well on the main sorting task (Figure 3.1), and also an auxiliary task. This auxiliary task corresponds to a simpler sub-task associated with "truly learning" a solution to the main task. In this section, let us focus on the *successor task*: given an input sequence and a particular element $a$ from the sequence, the model must learn to predict its successor, i.e., the element that follows $a$ in the sorted sequence (see Figure 3.4 for an example). .

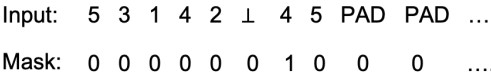

Figure 3.4: An example input sequence for the successor task.

In order to jointly learn the two tasks, we use the hard-parameter-sharing model for multi-task learning (Crawshaw, 2020) where the entire model backbone is shared across the two tasks, and a task-specific classification head is used at the final layer to make predictions for the respective tasks. We train the models as before for $100k$ steps, each time alternating between performing gradient updates on the main task and auxiliary task. The training dataset size is split equally among the two tasks.

---

[1]The in-distribution test accuracy always reaches $100\%$ well within the first $100k$ steps.

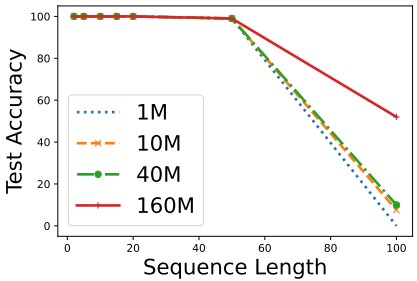

*Figure 3.5: Effect of data scaling for task hinting. We observe consistent improvements in test accuracy on higher length sequences.*

Figure 3.5 shows the effect of scaling the training-set size on a depth-2 model with task hinting. In contrast to the single-task setup, we see consistent gains as the training set size increases. In particular, for dataset size of $160M$ the test accuracy for length 100 reaches to $52.4\%$. Furthermore, by modifying the training set slightly so that $10\%$ of the sequences involve non-trivial repetitions (see Appendix D for details), the depth-2 model trained via task hinting achieves $92.6\%$ test accuracy on length 100 sequences! In contrast, the model obtained without task hinting continues to have test accuracy close to 0 on length 100 sequences, even on this modified training set.

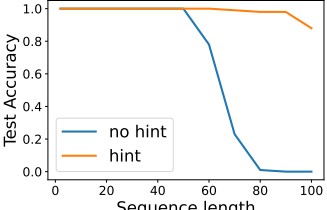

*Figure 3.6: Comparing test accuracy for hinting vs. no hinting for increasing sequence lengths; higher is better.*

*Figure 3.7: Comparing the edit distance for hinting vs. no hinting for increasing sequence lengths; lower is better.*

In Figures 3.6 and 3.7 we compare the test performance of the depth-2 model trained on a 160M dataset via task hinting and the model obtained via standard training, as we increase the test sequence length. Both the models are trained on the modified dataset that contains $10\%$ of sequences with non-trivial repetitions. We look at two metrics: (a) the *full-sequence* accuracy, i.e., whether the model outputs the entire sorted sequence correctly, and (b) the *edit distance* between the true sorted sequence and the predicted sequence. We see the performance of the no-hinting model drops sharply with sequence length; in contrast, the performance of the hinting model remains much more stable.

To further investigate the robustness of the trained models, we test them on distributions beyond uniform random sampling. We construct test distributions of the form $\text{rep}(i, r)$, where a sequence of length $i$ is created by sampling $\lfloor i/r \rfloor$ elements uniformly at random without replacement, and

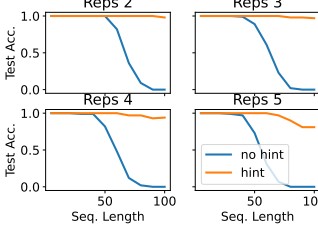

*Figure 3.8: Test accuracy comparison of hinting vs. no hinting models on repetitions.*

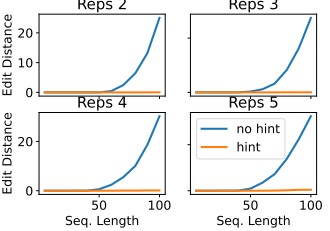

*Figure 3.9: Edit distance comparison of hinting vs. no hinting on repetitions.*

repeating each $r$ times. (The remaining $i - \lfloor i/r \rfloor r$ elements are drawn uniformly at random with replacement.) Figures 3.8 and 3.9 compare the performance of the hinting-based and no-hinting models for repetition values ($r$) in $\{2, 3, 4, 5\}$. Again, we observe that the hinting-based models are stable in their performance, both in terms of their full-sequence accuracy and their edit distance.

**Alternative Hints.** Many other natural auxiliary tasks can serve as hints for the principal task of sorting. In Figure 3.10 we present two such tasks. The first is a "*count*" task where, given a sequence of only two numbers repeated a certain number of times the model has to identify the least occurring one. The underlying idea is that sorting requires producing an output with the correct number of occurrences of any particular number, and hence understanding whether the output contains fewer or equal occurrences of a number. A very similar intuition underlies the second task, which is a "*fill*" task: given a sequence containing a single number repeated some number of times, followed by a prefix of that sequence, the model has to fill in the remaining entries.

| **Count Hint** | | | | | | | | |
|---|---|---|---|---|---|---|---|---|
| Input: | 5 5 5 4 5 | ⊥ | 4 | PAD | PAD | | | |
| Mask: | 0 0 0 0 0 | 1 | 0 | 0 | 0 | | | |

| **Fill Hint** | | | | | | | | | |
|---|---|---|---|---|---|---|---|---|---|
| Input: | 5 5 5 ⊥ 5 | ⊥ | 5 | 5 | PAD | PAD | .... |
| Mask: | 0 0 0 0 0 | 1 | 1 | 1 | 1 | 0 | .... |

*Figure 3.10: An example input sequence for count hints and fill hints.*

We now compare the performance of the models trained via the three different types of hints—the successor hint from the previous secton, and these count and fill hints—in Figure 3.11. Observe that the length generalization varies greatly depending on the type of hint used. In particular, while the fill hints result in a marginal improvement over the standard model without hinting, the use of count hints results in a worse performance than having no hints at all!

## 4 INTERPRETING HINTS

Given the large differences in the performance of models trained with different kinds of hints, we now turn to visualization and probing techniques to try and understand the mechanism by which the network learns the sorting task. To begin with, some notation:

1. For a given trained model, let $E \in \mathbb{R}^{q \times d}$ be the learned input embedding (usually called the *embedding table*); here $q$ is the vocabulary size, and $d$ is the embedding dimensionality. (In our experiments, $q = 103$ and $d = 1024$.) We call the rows of $E$ the *encoder basis*; the use of the term "basis" is not unreasonable here, since experimentally we find that the rows are nearly orthogonal and of very similar lengths.

2. Let $(W, b)$ denote the classifier used at the last layer to make the next-token prediction. Here $W$ is a $d \times q$ matrix (usually called the *softmax layer*), and $b$ is the *bias* vector of size $d$. We also observe that the columns of $W$ are nearly-orthogonal, and we call these vectors the *decoder basis*. These two bases are nearly orthogonal to each other as well, and hence span $2q = 206$ of the $d = 1024$ dimensions.

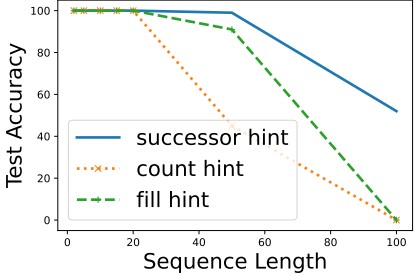

*Figure 3.11: Test accuracy comparison of various hinting tasks. Not all auxiliary tasks lead to improved length generalization, and some (such as counting) leads to performance degradation.*

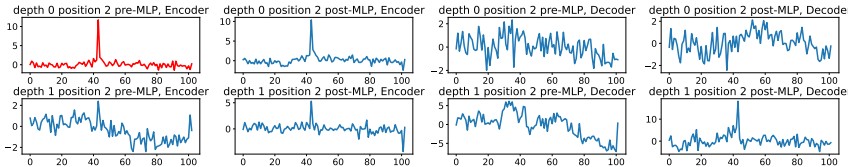

*Figure 4.12: The projection of token 43 at position 2 onto the encoder and the decoder bases. We observe a noisy copy operation being implemented in the encoder basis (see row 1, plot 1 in red).*

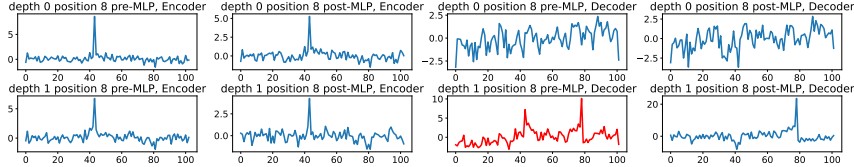

*Figure 4.13: The projection of token 43 at position 8 onto the encoder and the decoder bases. We observe an* Identity+Successor *operation being implemented in the decoder basis after the second attention layer (see row 2, plot 3 in red).*

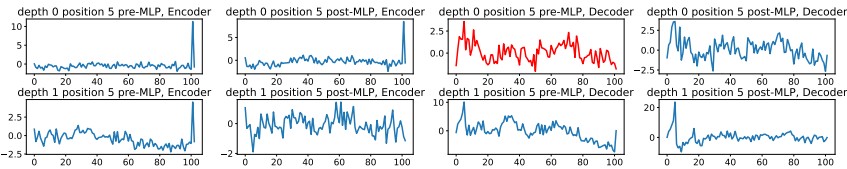

*Figure 4.14: The projection of token $\perp$ at position 5 onto the encoder and the decoder bases. We observe a noisy min operation being implemented in the decoder basis (see row 1, plot 3 in red).*

As the network performs inference on an input $\boldsymbol{\sigma} = \langle \sigma_0, \ldots, \sigma_{T-1} \rangle$, we can compute the intermediate embeddings for each token $\sigma_i$ in the sequence and visualize them in the encoder and decoder bases. Formally, a standard decoder-only transformer model consists of layers of *attention blocks*, where each attention block consists of a layer of *self-attention* followed by a layer of *MLP*. Hence, for a given input $\boldsymbol{\sigma}$ and position index $i$, let $X_{i,j}^{\text{pre}}$ denote the embedding of token $\sigma_i$ obtained at depth $j^{th}$ *before* applying the MLP at that depth, and let $X_{i,j}^{\text{post}}$ the embedding *after* applying the MLP. We then visualize several positions $i$ for various inputs by projecting the pre-MLP and the post-MLP embeddings onto the encoder and the decoder bases. These projections are often insightful, since the basis vectors naturally correspond to vocabulary symbols.

As an example consider the input sequence: $\boldsymbol{\sigma} = \langle 5, 17, 43, 78, 92, \perp \rangle$ of five numbers that have to be sorted. We consider a depth-two trained model (via standard training) and plot the projected embeddings for token $\sigma_2 = 43$ in Figure 4.12. The embeddings after first attention layer (depth-0 pre-MLP) are highly concentrated on the token 43 in the encoder basis, suggesting a (noisy) *copy* operation being implemented by the layer. This tendency of tokens to simply copy themselves in the encoder basis is observed for tokens appearing before the $\perp$ token at all points in the inference.

Next, in Figure 4.13 we plot the token 43 again, but now when it appears at position 8, i.e., when it is part of the output sequence. We again observe the noisy copy operation in the encoder basis, but the behavior in the decoder basis is quite different. Specifically, the embedding after the second attention layer (depth-1 pre-MLP) is highly concentrated on both token 43 *and on its successor* in the sorted sequence, i.e., on token 78. In fact, we consistently observe this two-peak phenomenon in the depth-1 pre-MLP embedding for tokens in the output sequence—they appear to implement an *Identity+Successor* operation. The final MLP layer then acts as a *denoiser*, reducing/removing the spike on the identity part to ensure that the final embeddings are concentrated correctly on the successor element—hence the classification based on $(W, b)$ correctly outputs the successor element.

Finally, let us examine the embeddings for the end-of-input $\perp$ token in Figure 4.14. Here we consistently observe that a noisy *minimum operation* is being implemented right after the first attention layer (depth-0): the embedding has largest inner product with the vector in the decoding basis that corresponds to the minimum element in the input!

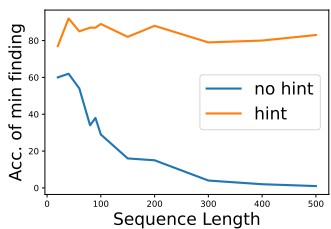

*Figure 4.15: The accuracy of implementing the min finding operation after layer-1 attention.*

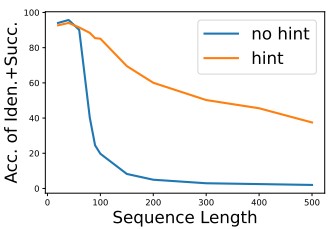

*Figure 4.16: The accuracy of implementing the Identity+Successor operation after layer-2 attention.*

To summarize, consider input $\sigma_0, \ldots, \sigma_{n-1}$, with the $\perp$ token at location $n$. We consistently observe that the embeddings suggest the following learning mechanism:

(i) Any token $\sigma_i$ in position $i < n$ has a sharp spike on the encoding basis vector corresponding to symbol $\sigma_i$ throughout the inference.

(ii) The embedding for the end-of-input delimiter $\perp$ typically implements a noisy minimum operation in the decoding basis after the depth-0 self-attention later.

(iii) Any token in position $i > n$ (i.e., part of the output) often implements the *Identity+Successor* operation after the depth-1 self-attention layer. The depth-1 MLP acts as a *denoiser*, removing the spike on the symbol itself, which then correctly highlights only the successor.

The empirical evidence suggests that the network aims to solve the sorting task using a natural algorithm: (a) first finding the minimum element to follow the $\perp$ symbol, and thereafter (b) computing the successor element for each element. Moreover, this suggests why the successor hints are highly beneficial: these hints align well with the solution concepts that the network is trying to learn. In order to further validate this hypothesis we compare how effective the internal representations of these depth-2 models (trained with/without hints) are at implementing the above-mentioned mechanisms. In particular, we measure how often:

(i) the embedding for the $\perp$ token after the depth-0 self-attention layer computes the minimum input element (this is measured by computing the dot-product of the embedding with the decoding basis), and

(ii) the embedding for tokens in the output sequence (those after $\perp$) correctly implement the *Identity+Successor* mechanism after the layer-2 attention operation.

Figures 4.15 and 4.16 show that using successor hints significantly improves the accuracy of these two mechanisms in the internal representations, especially at lengths not seen during training. We conjecture that in general, auxiliary tasks that align well with the implicit bias of the network tend to help the most to obtain out-of-distribution robustness.

Our analysis above shows that direct projection-based techniques can help demystify some algorithmic mechanisms underlying transformer networks, and provide interesting insights. Moreover, the generality of the techniques gives hope that they can be used for other large-scale problems.

## 5 THEORETICAL ANALYSIS

The previous sections relied on the toolkit of visualization and probing using the encoder/decoder bases to gather empirical evidence about the learned mechanism, and the effectiveness of the successor finding task. In this section we ask the questions: *can we give a theoretical construction that matches the empirical findings, and that can be implemented via a shallow transformer model? What does this construction tell us about length generalization?* Recent theoretical works have alluded to the possibility that log-precision transformers may capture the complexity class of $TC^0$ circuits (Merrill et al., 2022). Since (Chandra et al., 1984) show that sorting is indeed in $TC^0$, it is conceivable that one can design constant-depth transformer models for sorting.

While there may be many such constructions of shallow transformer models, we impose some additional constraints: (a) we ask for a depth-two model, and (b) the size of the network should be

independent of the input length $n$, even though the parameters could depend logarithmically on $n$. Finally, we want a construction that displays the empirical properties we observe in Section 4. We hope that by getting a theoretical construction that is close to the empirically observed behavior, we may be able to generate more practically useful insights from the theory.

Formally, we fix an alphabet $\Sigma$ of size $q$. We have one special symbol $\perp$, which is the end-of-sequence delimiter. Let $\Sigma'$ denote the extended alphabet $\Sigma \cup \{\perp\}$. We associate $\Sigma$ with the naturals $\{1, 2, \ldots, q\}$, with the usual total order on them. Since we seek to sort sequences using next-token prediction, the input is a sequence of length $T$ consisting of three conceptual parts:

1. pre-delimiter: a sequence $\sigma_0, \sigma_1, \ldots, \sigma_{n-1}$ where each $\sigma_i \in \Sigma$. These represent the unsorted input.
2. the end-of-sequence delimiter: $\sigma_n = \perp$.
3. post-delimiter: a sequence of $i = T - n - 1$ symbols $\sigma_{n+1}, \sigma_{T+2}, \ldots, \sigma_{T-1}$ from $\Sigma$, which ideally represent the smallest $i$ symbols in the input $\boldsymbol{\sigma}_{[0:n-1]}$ (in non-decreasing order).

Given this sequence $\boldsymbol{\sigma}$ we want to predict the next symbol in the sorted order of the input $\boldsymbol{\sigma}_{[0:n-1]}$. Finally, we consider transformers with the tempered softmax operation, i.e., given $x \in \mathbb{R}^d, \text{softmax}_\tau(x)_i = e^{\tau x_i} / \sum_j e^{\tau x_j}$. In our construction, we consider transformer models where $\tau = \beta \ln n$ and $\beta$ is a tunable/learnable parameter, and $n$ is the sequence length. This is a departure from the standard practice of always setting $\tau = 1$, *independent of the input length*. We prove the following theorem:

**Theorem 5.1.** *For any alphabet of size $q$ and bit precision complexity $b$, there exists a depth-2 decoder only transformer model with two attention heads, embedding dimensionality and hidden layer dimensionality of $O(q)$, and network weights encoded using $b$ bits of precision that correctly solves the sorting task on sequence of length up to $2^{\Omega(b)}$. Furthermore, the network displays the following characteristics:*

1. *For any position $i < n$, the embedding obtained after the first attention layer is highly concentrated on $\sigma_i$ in the encoding basis, hence implementing a* copy *operation.*
2. *For token $\perp$, the embedding after the first attention layer has the highest dot product (in the decoding basis) with the smallest element in the sequence, hence implementing the* min *operation.*
3. *For any position $i > n$, the embedding obtained after the second attention layer is concentrated (in the decoding basis) on the token at position $i$ and the next largest element in the sorted sequence, thereby implementing the* Identity&Successor *operation.*

**Algorithmic Implications.** Note that our theoretical construction relies on the ability to apply length-dependent tempered-softmax operations. This is important for us to ensure that the performance of the network does not degrade with increasing sequence lengths. Given this theoretical construction, we ask whether incorporating length-dependent tempered-softmax operations suggested by the theory could help with length generalization in practice. In order to implement this, we modify the `Flaxformer` codebase (Heek et al., 2023) to introduce the tempered softmax at each attention layer (with each its own *learnable* $\beta$ parameter). We train depth-two transformer models on the same training set of size 160M, both with and without hints, and compare the performance with and without tempered softmax operations.

|  | Standard softmax | Tempered softmax |
|---|---|---|
| Test Acc. 50 | 89 | **99.4** |
| Test Acc. 100 | 0.0 | **45.2** |

*Table 1: Test accuracy of depth-2 model trained without hints and with/without tempered softmax.*

|  | Standard softmax | Tempered softmax |
|---|---|---|
| Test Acc. 50 | 99.2 | **99.7** |
| Test Acc. 100 | 52.4 | **64.8** |

*Table 2: Test accuracy of depth-2 model trained with successor hints and with/without tempered softmax.*

As we observe in Tables 1 and 2, the introduction of the tempered softmax significantly improves length generalization of models trained via standard training, as well as those trained via task hinting. Furthermore, the tempered softmax helps across all data scale ranges. In particular, even for the model trained without hints on a training set of size 1M, the test accuracy on sequences of length 100 increases from $0\%$ to $42\%$ due to the introduction of the tempered softmax!

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

## A  DISCUSSION AND LIMITATIONS

In this work we proposed *task hinting* as an effective approach for the problem of length generalization. We observe that using hints that have a strong alignment with the internal biases of the learning mechanism can result in significant gains in out-of-distribution robustness for the problem of sorting integers. For this setting, we use probing and visualization-based techniques to investigate the internal learning mechanisms; these allow us to explain the success of the successor-based hints that we use in our experiments. In general, even these probing/visualization approaches may not always be feasible for large-scale settings, so designing the appropriate hinting tasks may be a problem in itself: it would be good to develop a principled approach for deciding on hinting tasks.

While we observed that other natural hinting tasks, such as the count task and the fill task did not help (and sometimes even hurt the performance), we feel that these are useful auxiliary capabilities for a sorting network, and it would be good to understand their lack of success at a deeper level. Moreover, it would also be interesting to combine multiple hints, and make the network benefit from learn more than two tasks simultaneously. We tried this approach for the sorting problem, where we trained the model to do well on all the three types of hinting tasks simultaneously, but observed mixed or even negative results.

Our work also proposes the introduction of length-dependent parameters into the attention mechanism, and observe that they significantly boost the robustness of the models for both the sorting problem and the increment problem. It would be interesting to apply this to larger-scale settings of training language models, and to evaluate whether any gains in robustness can be obtained on more general reasoning tasks. Finally, when using the framework of multitask learning to make the network learn both tasks simultaneously, we did not make efforts to optimize the various parameters of the setup, and followed a simple recipe of alternating gradient updates on each task. Further optimizations in this stage could lead to better performance.

## B  FORMAL CONSTRUCTIONS [PROOF OF THEOREM 5.1]

We now show how to implement a min/successor operation via two-layer transformers, which allows us to sort using next-token predictions.

### B.1  NOTATION

Fix an alphabet $\Sigma$ of size $q$. We have one special symbol $\perp$, which is the end-of-sequence delimiter. Let $\Sigma'$ denote the extended alphabet $\Sigma \cup \{\perp\}$. We associate $\Sigma$ with the naturals $\{1, 2, \ldots, q\}$, with the usual total order on them.

Since we seek to sort sequences using next-token prediction, the input is a sequence of length $T$ consists of three conceptual parts:

1. the pre-delimiter part: a sequence of $n$ symbols $\sigma_0, \sigma_1, \ldots, \sigma_{n-1}$ where each $\sigma_i \in \Sigma$. These represent the unsorted input.

2. the end-of-sequence delimiter: $\sigma_n = \perp$.

3. the post-delimiter part: a sequence of some $i = T - n - 1$ symbols $\sigma_{n+1}, \sigma_{T+2}, \ldots, \sigma_{T-1}$ again from $\Sigma$, which ideally represent the smallest $i$ symbols in the input $\boldsymbol{\sigma}_{[0:n-1]}$ (in non-decreasing order).

Given this sequence $\boldsymbol{\sigma}$ we want to predict the next symbol in the sorted order of the input $\boldsymbol{\sigma}_{[0:n-1]}$.

### B.2  THE TRANSFORMER ARCHITECTURE

The process works as follows:

1. The initial *embedding function* $\mathsf{Enc} : \Sigma' \to \mathbb{R}^d$ maps each symbol to a vector in $\mathbb{R}^d$. Let $\mathbf{X}_0 \in \mathbb{R}^{T \times d}$ be the embedding of the input sequence $\boldsymbol{\sigma}$, where the $i^{th}$ row of $\mathbf{X}_0$ equals $\mathsf{Enc}(\sigma_i)$. We use $\mathbf{X}_{ti}$ to denote the $i^{th}$ row of $\mathbf{X}_t$.

2. There are $b$ *attention blocks* which transform this input: we denote the operation of attention block $t$ by $B_t : \mathbb{R}^{T \times d} \to \mathbb{R}^{T \times d}$, and hence

$$\mathbf{X}_t := B_t(\mathbf{X}_{t-1}).$$

Each block contains a *self-attention layer* and *a multi-layer perceptron (MLP) layer*, followed a *layer normalization* operation, such that

$$B_t := \left( \text{normalize} \circ (I + f_t^{mlp}) \circ (I + f_t^{attn}) \right).$$

The identity maps are the *residual* stream to which the results of the various operations get repeatedly added in.

3. Each *self-attention layer* consists of $h$ *attention heads*: the columns of the matrix $\mathbf{X}_t$ are split into $h$ matrices $\mathbf{X}_{t1}, \ldots, \mathbf{X}_{th}$, each with $d/h$ columns and $T$ rows. Each head is specified by matrices $Q, K, V$. It takes a matrix $\overline{X} \in \mathbb{R}^{T \times d/h}$ and produces a matrix of the same dimensions as follows:

$$f^{attn}(\overline{\mathbf{X}}; K, Q, V) := \text{smax}_\tau (\overline{\mathbf{X}} K Q^\intercal \overline{\mathbf{X}}^\intercal) \overline{\mathbf{X}} V$$

Here the smax operator takes a matrix $A$ and a parameter $\tau$ and defines

$$\text{smax}_\tau(A)_{ij} = \frac{e^{\tau A_{ij}} \mathbf{1}_{(i \geq j)}}{\sum_{j' \leq j} e^{\tau A_{ij'}}}.$$

(Note that the above operator combines the softmax operation and the auto-regressive behavior.) Finally, the resulting $h$ sub-matrices are concatenated together to give the result of the entire self-attention layer; let $f_t^{attn}$ denote the composite function. Define $\mathbf{Y}_t := (I + f_t^{attn}) \mathbf{X}_{t-1}$ as the result of adding back the original signal to the result.

4. Next, the *multi-layer perceptron* (MLP) layer (which in our case is a two-layer perceptron) is a transformation $f^{mlp}(\mathbf{Y}; W_1, W_2, b_1, b_2)$ that is specified by two matrices $W_1, W_2$ and bias vectors $b_1, b_2$. It is the result of applying the following map to each row $y^\intercal$ of $\mathbf{Y}$ separately:

$$y \mapsto W_2 \, \sigma(W_1 y + b_1) + b_2.$$

Here the map $\sigma(\cdot)$ is usually the component-wise ReLU operation (or in more complicated settings, other non-linear operators like GeLU or GLU).

5. The final piece in each block is the *layer normalization* operation, which again is applied to each row of the current embedding independently. Given a vector $x \in \mathbb{R}^d$, it subtracts $\mu := \|x\|_1 / d$ from each coordinate to make it zero-mean, and then divides each entry by $\sigma := \sqrt{\sum_i x_i^2 / d}$; this makes the Euclidean length $\sqrt{d}$. We denote this operation by normalize.

6. Unrolling, the entire transformer map is

$$\mathbf{X}_b = \left( B_b \circ B_{b-1} \circ \cdots \circ B_1 \right) \mathbf{X}_0.$$

7. The final transformation is the decoding/unembedding operation, which takes $\mathbf{X}_b \in \mathbb{R}^{T \times d}$ and applies some decoding map $\text{Dec} : \mathbb{R}^d \to \Sigma'$ independently on each row of $\mathbf{X}_b$. This produces characters in $\Sigma'$—these are the *predictions* for the next symbols. For our decoder-only constructions, the only relevant prediction is that of the last symbol: we output this prediction $\text{Dec}(\mathbf{X}_{b,T-1})$ as the next token, thereby increasing the length by 1—this longer string is then the input for the next iteration.

We now show how to implement each of these attention blocks for the sorting network.

## B.3 The Encoding Function

Fix a set of unit vectors $\{\mathbf{e}_s, \mathbf{e}'_s\}_{s \in \Sigma'}$ which are all orthogonal to each other in $\mathbb{R}^d$. The initial embedding is simple: each symbol $a \in \Sigma$ is encoded by the vector $\mathbf{e}_a + \mathbf{e}'_a$, and the end-of-input delimiter is encoded as $\mathbf{e}_\perp + \mathbf{e}'_\perp$. This gives us the input embedding $\mathbf{X}_0$.

### B.4    BLOCK #1

The first block has two goals: (i) it gets each token to implement a "min/copy" operation (in which the end-of-input delimiter predicts the minimum element from the input, whereas each other token just predicts itself), and (ii) the tokens corresponding to the same symbol before and after the end-of-input delimiter distinguish themselves, so that the second block can act on them accordingly.

#### B.4.1    BLOCK #1: SELF-ATTENTION LAYER

There are two attention heads in the first self-attention layer, each getting some $d/2$ columns of the matrix $\mathbf{X}_0$. We denote the resulting two sub-matrices by $\mathbf{X}_{01}, \mathbf{X}_{02} \in \mathbb{R}^{T \times d/2}$, and ensure that for each $s$, the span of $\{\mathbf{e}_s\}_{s \in \Sigma'}$ lies in the subspace corresponding to the first $d/2$ coordinates, and the span of $\{\mathbf{e}'_s\}$ lies in the one for the other $d/2$ coordinates.

In the entire construction, we set $\tau = 3 \ln n$, where $n$ is the length of the input. Define the $Q, K, V$ matrices for the attention heads as follows:

- Attention Head #1, which operates on a subspace containing the vectors $\{\mathbf{e}_0, \mathbf{e}_1, \mathbf{e}_2, \ldots, \mathbf{e}_q\}$: for some positive scalar $C \geq 1$ to be specified below, define

$$Q\mathbf{e}_a = \mathbf{e}_a + C\mathbf{e}_\perp \qquad K\mathbf{e}_a = \mathbf{e}_a \qquad V\mathbf{e}_a = \widetilde{\mathbf{e}}_a \qquad \text{(B.1)}$$
$$Q\mathbf{e}_\perp = \mathbf{e}_\perp \qquad K\mathbf{e}_\perp = \mathbf{e}_\perp \qquad V\mathbf{e}_\perp = \widetilde{\mathbf{e}}_\perp. \qquad \text{(B.2)}$$

  (Here, and subsequently, the matrices $Q, K, V$ map all vectors orthogonal to the specified vectors to zero.) The vectors $\{\widetilde{\mathbf{e}}_s\}_{s \in \Sigma'}$ are fresh orthonormal vectors.

- Attention Head #2, which operates on a subspace containing the vectors $\{\mathbf{e}'_0, \mathbf{e}'_1, \mathbf{e}'_2, \ldots, \mathbf{e}'_q\}$: define

$$Q\mathbf{e}'_a = \mathbf{e}'_a \qquad K\mathbf{e}'_a = \mathbf{e}'_a \qquad V\mathbf{e}'_a = \widehat{\mathbf{e}}'_a \qquad \text{(B.3)}$$
$$Q\mathbf{e}'_\perp = \sum_{b \in \Sigma} \gamma_b\, \mathbf{e}'_b \qquad K\mathbf{e}'_\perp = \mathbf{e}'_\perp \qquad V\mathbf{e}_\perp = 0. \qquad \text{(B.4)}$$

  Here $\{\gamma_b\}_{b \in \Sigma}$ are also values to be specified soon. Again, the vectors $\{\widehat{\mathbf{e}}'_b\}_{b \in \Sigma}$ are fresh orthonormal vectors.

This means that for any symbol $a \in \Sigma$ at some position $i$ before the $\perp$ delimiter, the first attention head outputs

$$\frac{\sum_{j \leq i : \sigma_j = a} e^\tau\, \widetilde{\mathbf{e}}_{\sigma_j} + \sum_{j \leq i : \sigma_j \neq a} \widetilde{\mathbf{e}}_{\sigma_j}}{\sum_{j \leq i : \sigma_j = a} e^\tau + \sum_{j \leq i : \sigma_j \neq a} 1} = \frac{n_{a,[0,i]}\, e^\tau\, \widetilde{\mathbf{e}}_a + \sum_{j \leq i : \sigma_j \neq a} \widetilde{\mathbf{e}}_{\sigma_j}}{n_{a,[0,i]} e^\tau + (i + 1 - n_{a,[0,i]})}$$

Here $n_{a,[x,y]}$ is the number of occurrences of $a$ in the multiset $\{\sigma_x, \ldots, \sigma_y\}$. Now since $\tau \geq 3 \ln n$, most of the attention is on all occurrences of the same symbol $a$ seen thus far, and hence this expression is

$$(1 - O(1/n^4)) \cdot \widetilde{\mathbf{e}}_a + O(1/n^2) \cdot \mathbf{u}_{1i},$$

where $\mathbf{u}_1$ is some "error" vector of unit norm. Similarly, the second attention head gives

$$(1 - O(1/n^4)) \cdot \widehat{\mathbf{e}}_a + O(1/n^2) \cdot \mathbf{u}_{2i},$$

for some other error vector $u_2$. Hence, adding back in the residual, we get that the $i^{th}$ entry (for $i < n$, where $\sigma_i = a$ for some $a \in \Sigma$) gives us

$$(I + f_1^{attn})(\mathbf{X}_{0i}) \approx \mathbf{X}_{0i} + \widetilde{\mathbf{e}}_a + \widehat{\mathbf{e}}_a = \mathbf{e}_a + \mathbf{e}'_a + \widetilde{\mathbf{e}}_a + \widehat{\mathbf{e}}_a. \qquad \text{(B.5)}$$

Here and henceforth, we will use the "$\approx$" to hide error vectors of length $O(1/n^2)$.

Now a similar analysis shows that for position $i > n$ (such that $\mathbf{X}_{0i} = a$), setting $C = 3$ then most of $a$'s attention (in the first head) is on the $\perp$ delimiter, and hence

$$(I + f_1^{attn})(\mathbf{X}_{0i}) \approx (\mathbf{e}_a + \mathbf{e}'_a) + \widetilde{\mathbf{e}}_\perp + \widehat{\mathbf{e}}'_a. \qquad \text{(B.6)}$$

Finally, the $\perp$ delimiter pays most of its attention to itself in first head, whereas in the second head it pays attention to all the tokens (weighted by the $e^{\tau\gamma_b}$ multipliers). Defining $\alpha_b := e^{\tau\gamma_b}$, we get

$$(I + f_1^{attn})(\mathbf{X}_{0i}) \approx (\mathbf{e}_\perp + \mathbf{e}'_\perp) + \widetilde{\mathbf{e}}_\perp + \frac{\sum_b \alpha_b n_{b,[0,n)} \widehat{\mathbf{e}}'_b}{\sum_b \alpha_b n_{b,[0,n)} + 1}. \tag{B.7}$$

Since we have identified the symbols of $\Sigma$ with $\{1, 2, \ldots, q\}$, we can define $\gamma_b = (q - b + 1)$, and hence $\ln \alpha_b := 3(q - b + 1) \ln n$. Since the $\alpha_b$ values decrease rapidly as $b$ increases, the fraction on the right assigns most of its weight to vector $\widehat{\mathbf{e}}'_b$ corresponding to the minimum element in the input $\sigma_{[0,n)}$. In other words, we get

$$(I + f_1^{attn})(\mathbf{X}_{0i}) \approx (\mathbf{e}_\perp + \mathbf{e}'_\perp) + \widetilde{\mathbf{e}}_\perp + \min_{b \in \boldsymbol{\sigma}_{[0,n-1]}} \widehat{\mathbf{e}}'_b. \tag{B.8}$$

Let us denote the output of the first self-attention layer by $\mathbf{Y}_1$; i.e.,

$$\mathbf{Y}_1 := (I + f_1^{attn})(\mathbf{X}_0).$$

### B.4.2 BLOCK #1: MLP LAYER

Recall that the MLP layer is applied to each embedding separately, and there is no interaction between the embeddings of different tokens. The first MLP layer has two goals:

1. The first goal is to convert the $\widetilde{\mathbf{e}}_\perp$ vector in embedding of some post-delimiter $a$ to the corresponding $-\widetilde{\mathbf{e}}_a$. To this end, the

$$f_{1,1}^{mlp}(x) := \sum_{b \in \Sigma} \sigma(\langle x, \mathbf{e}_b \rangle + \langle x, \widetilde{\mathbf{e}}_\perp \rangle - 1)(-\widetilde{\mathbf{e}}'_b - \widetilde{\mathbf{e}}_\perp). \tag{B.9}$$

   Recall that $\sigma(z) := \max(0, z)$ is the ReLU function.

2. The second goal is to shift the coordinates of the $\widetilde{\mathbf{e}}_a$ vectors, so that they appear in the second half of the coordinates instead of the first. For this we use

$$f_{1,2}^{mlp}(x) := \sum_{b \in \Sigma} \left( (\sigma(\langle x, \widetilde{\mathbf{e}}_b \rangle) - \sigma(\langle x, -\widetilde{\mathbf{e}}_b \rangle)) \cdot (\widetilde{\mathbf{e}}'_b - \widetilde{\mathbf{e}}_b) \right) \tag{B.10}$$

Finally, $f_1^{mlp}(x) := f_{1,1}^{mlp}(x) + f_{1,2}^{mlp}(x)$. This gives us the output of the first attention block:

$$\mathbf{X}_1 := (I + f_1^{mlp})(\mathbf{Y}_1).$$

As mentioned above, we do not use the layer normalization in this construction, so this $\mathbf{X}_1$ is now fed to the second attention block.

To summarize,

$$\mathbf{X}_{1i} = (I + f_1^{mlp})(\mathbf{Y}_{1i}) = \begin{cases} (\mathbf{e}_a + \mathbf{e}'_a) + \widetilde{\mathbf{e}}'_a + \widehat{\mathbf{e}}'_a + \mathbf{u}_{3i} & \text{for } i < n \\ (\mathbf{e}_a + \mathbf{e}'_a) - \widetilde{\mathbf{e}}'_a + \widehat{\mathbf{e}}'_a + \mathbf{u}_{3i} & \text{for } i > n, \text{ and} \\ (\mathbf{e}_\perp + \mathbf{e}'_\perp) + \widetilde{\mathbf{e}}_\perp + \min_{b \in \boldsymbol{\sigma}_{[0,n-1]}} \widehat{\mathbf{e}}'_b + \mathbf{u}_{3i} & \text{for } i = n. \end{cases}$$

The error vectors $\mathbf{u}_{3i}$ above have magnitude $O(1/n^2)$.

### B.5 BLOCK #2

The second block now ensures that the $\perp$ token predicts the minimum element, whereas each other token predicts its successor. The non-trivial part of this construction arises from duplicates in the input, so that each symbol $a \in \Sigma$ has to infer whether the number of copies of $a$ already output equals the number in the input part of $\boldsymbol{\sigma}$, and accordingly predict whether to output another $a$ or the successor to $a$. (Observe that this is an *ordinal* concept, and not a *cardinal* one: the network does not need the actual count of the $a$'s that have been output, but to just know whether the number of $a$'s output is strictly less than the number in the input.)

### B.5.1 BLOCK #2: SELF-ATTENTION LAYER

The self-attention layer of the second block again has two attention heads:

- Attention Head #1, which again operates on a subspace containing the "unprimed" vectors:

$$Q\mathbf{e}_a = \mathbf{e}_a \qquad K\mathbf{e}_a = \mathbf{e}_a \qquad V\widetilde{\mathbf{e}}_a = \widehat{\mathbf{e}}_a \qquad (\text{B.11})$$
$$Q\mathbf{e}_\perp = \mathbf{e}_\perp \qquad K\mathbf{e}_\perp = \mathbf{e}_\perp \qquad V\mathbf{e}_\perp = 0. \qquad (\text{B.12})$$

Again, the matrices $Q, K, V$ map all vectors orthogonal to the specified vectors to zero. Recall that the $\tau$ parameter is the softmax operator is set to $3\ln n$.

- Attention Head #2, which operates on the primed vectors: define

$$Q\mathbf{e}'_a = \sum_{b>a}\gamma_b\mathbf{e}'_b \qquad K\mathbf{e}'_a = \mathbf{e}'_a \qquad V\mathbf{e}'_a = \varepsilon\widehat{\mathbf{e}}'_a \qquad (\text{B.13})$$
$$Q\mathbf{e}'_\perp = \mathbf{e}'_\perp \qquad K\mathbf{e}'_\perp = \mathbf{e}'_\perp \qquad V\mathbf{e}'_\perp = 0. \qquad (\text{B.14})$$

(We will fix the value of $\varepsilon > 0$ below.)

Since we are at the final block, we are no longer concerned with the part of the input in $\sigma_{[0:n-1]}$, and hence focus on positions $n$ and beyond. The $\perp$ delimiter at position $n$ primarily pays attention to itself in both attention heads, since $\tau$ is $\Omega(\log n)$. This means it remains unchanged, and

$$(I + f_2^{attn})(\mathbf{X}_{1n}) = (\mathbf{e}_\perp + \mathbf{e}'_\perp) + \widetilde{\mathbf{e}}_\perp + \min_{b\in\boldsymbol{\sigma}_{[0,n-1]}}\widehat{\mathbf{e}}'_b + \mathbf{u}_{4n}, \qquad (\text{B.15})$$

where the new error vector $\mathbf{u}_{4n}$ is still of the order $O(1/n^2)$.

Next, consider any position $i > n$, such that $\sigma_i = a$ for some $a \in \Sigma$. The first attention head gives

$$\frac{\sum_{j\leq i:\sigma_j=a}e^C\,V(\mathbf{X}_{1j}) + \sum_{j\leq i:\sigma_j\neq a}V(\mathbf{X}_{1j})}{e^C\,n_{a,[0.i]} + (n - n_{a,[0,i]})} = \frac{e^C(n_{a,[0,n]} - n_{a,[n+1,i]})\,\widehat{\mathbf{e}}_a + \sum_{j\leq i:\sigma_j\neq a}\widehat{\mathbf{e}}_{\sigma_j}}{e^C\,n_{a,[0.i]} + (n - n_{a,[0,i]})} \qquad (\text{B.16})$$

Again, since $C = \Omega(\ln n)$, this is approximately

$$(n_{a,[0,n]} - n_{a,[n+1,i]})\,\widehat{\mathbf{e}}_a + \mathbf{u}_{4i}, \qquad (\text{B.17})$$

where the error vector $\mathbf{u}_{4i}$ has tiny norm $O(1/n^2)$. The second attention head for the same symbol $\sigma_i$ gives

$$\frac{\sum_{b>a}\alpha_b n_{b,[0,i]}\,\varepsilon\,\widehat{\mathbf{e}}'_b + \sum_{b\leq a}n_{b,[0,i]}\,\varepsilon\,\widehat{\mathbf{e}}'_b}{\sum_{b>a}\alpha_b n_{b,[0,i]} + \sum_{b\leq a}n_{b,[0,i]}}. \qquad (\text{B.18})$$

Recall that $\alpha_b = e^{\tau\gamma_b} = n^{3(q-b+1)}$. If the symbol $a$ is not the largest symbol of the input (so that other symbols $b > a$ follow it in the input), this expression is $\varepsilon(\min_{b>a}\widehat{\mathbf{e}}'_b + \mathbf{u}'_{4i})$, with the error vector $\mathbf{u}'_{4i}$ having a tiny norm compared to $\min_{b>a}\widehat{\mathbf{e}}'_b$. As before, we define

$$\mathbf{Y}_2 := (I + f_2^{attn})(\mathbf{X}_1)$$

to be the outcome of this self-attention layer.

Let $\widehat{P}$ be the projection of these embeddings on the subspace spanned by the "hatted" vectors $\{\widehat{\mathbf{e}}_a, \widehat{\mathbf{e}}'_a\}_{a\in\Sigma}$. Then

$$\widehat{P}\mathbf{Y}_{2i} = \widehat{P}(I + f_2^{attn})(\mathbf{Y}_{1i})$$
$$= \begin{cases} \min_{b\in\boldsymbol{\sigma}_{[0,n-1]}}\widehat{\mathbf{e}}'_b + \mathbf{u}_{5i} & \text{for } i = n, \text{ and} \\ (n_{a,[0,n]} - n_{a,[n+1,i]})\,\widehat{\mathbf{e}}_a + \widehat{\mathbf{e}}'_a + \varepsilon\frac{\sum_{b>a}\alpha_b n_{b,[0,i]}\,\widehat{\mathbf{e}}'_b + \sum_{b\leq a}n_{b,[0,i]}\,\widehat{\mathbf{e}}'_b}{\sum_{b>a}\alpha_b n_{b,[0,i]} + \sum_{b\leq a}n_{b,[0,i]}} + \mathbf{u}_{5i} & \text{for } i > n. \end{cases} \qquad (\text{B.19})$$

### B.5.2 BLOCK #2: MLP LAYER

The final MLP layer of the second and final block has a simple task:

$$f_2^{mlp}(x) := \sum_{b \in \Sigma} \sigma(\langle x, -\widetilde{\mathbf{e}}_b \rangle) \cdot (-\widehat{\mathbf{e}}_b'). \tag{B.20}$$

This has the effect of adding in $(-\widehat{\mathbf{e}}_a')$ to any post-delimiter $a$, and hence "nullifying" the $\widehat{\mathbf{e}}_a'$. The net effect (again seen after projection onto the hatted subspace) is

$$\widehat{P}\mathbf{X}_{2i} = \widehat{P}(I + f_2^{mlp})(\mathbf{Y}_{2i})$$
$$= \begin{cases} \min_{b \in \boldsymbol{\sigma}_{[0,n-1]}} \widehat{\mathbf{e}}_b' + \mathbf{u}_{5i} & \text{for } i = n, \text{ and} \\ \left(n_{a,[0,n]} - n_{a,[n+1,i]}\right)\widehat{\mathbf{e}}_a + \varepsilon \frac{\sum_{b>a} \alpha_b n_{b,[0,i]} \widehat{\mathbf{e}}_b' + \sum_{b \le a} n_{b,[0,i]} \widehat{\mathbf{e}}_b'}{\sum_{b>a} \alpha_b n_{b,[0,i]} + \sum_{b \le a} n_{b,[0,i]}} + \mathbf{u}_{5i} & \text{for } i > n. \end{cases}$$

### B.6 THE DECODING LAYER

*Proof of Theorem 5.1.* The decoding (or unembedding) layer outputs the element $a$ for which the vector $\widehat{\mathbf{e}}_a + \widehat{\mathbf{e}}_a'$ has the largest inner product with the current embedding. In other words, $\sigma_i$ predicts

$$\arg\max_{a \in \Sigma} \quad \langle \mathbf{X}_{2i}, \widehat{\mathbf{e}}_a + \widehat{\mathbf{e}}_a' \rangle. \tag{B.21}$$

From the above construction we have the following properties that establish the correctness of the network:

1. For the delimiter at position $i = n$, this is simply the minimum element from $\boldsymbol{\sigma}_{[0,n-1]}$.

2. For any other location $i > n$ with $\sigma_i = a$, there are two cases:

   (a) Suppose there are multiple copies of $a$ in the input $\boldsymbol{\sigma}_{[0,n-1]}$, and not all of them have been output yet. This means $n_{a,[0,n]} > n_{a,[n+1,i]}$, and hence the maximum in (B.21) is achieved by $a$ itself, as long as $\varepsilon \le 1/2$, say. This results in predicting and outputting another copy of $a$.

   (b) Else suppose the number of copies of $a$ in the output already equals that in the input. In this case, the argmax in (B.21) is achieved at the smallest element $b \in \boldsymbol{\sigma}_{[0,n-1]}$ that is larger than $a$; this is indeed the correct "successor" element for $a$ to predict. One exception is when $i = 2n$, but then we do not need any further predictions.

Here we have crucially used that the maximizing vector has norm at least a constant, which means that the error vectors of length $O(1/n^2)$ do not alter the result. □

### B.7 THE LAYER NORMALIZATION

The construction above (using $d = O(|\Sigma|)$ coordinates) did not use the layer normalization operation; however, we can convert it to incorporate this operation as well. Recall that layer normalization operates on the embedding of each token independently: (i) given a vector $x \in \mathbb{R}^d$, it subtracts the mean $\mu_x := \frac{1}{d}\|x\|_1$ from each coordinate, and then (ii) renormalizes it to have squared norm $d$.

We take the above construction using vectors $x \in \mathbb{R}^d$ and extend it by adding $d$ new coordinates and appending an analogous construction using the negative of these vectors. The new embedding $\bar{x}$ has mean $\mu_{\bar{x}} = 0$, and hence the step (i) of layer normalization does not change anything.

This means that at the end of the first block, each of the embeddings in our construction have squared length $\approx 4$. This means the renormalization only changes the magnitude of the embeddings, but their relative sizes remain the same. Consequently, the computations in the second block remain unchanged. The final layer normalization again shifts and renormalizes the embedding, but this does not change the outcome.

### B.8 Agreement with Experimental Results

1. Consider the embedding after the first self-attention layer: decoding this embedding of $\perp$ (given in (B.7) gives us the minimum element, whereas decoding the embedding of any $\sigma_i$ for $i > n$ (as given in (B.6)) gives us the element $\sigma_i$ itself. This "min/copy" behavior can be observed in the experimental results.

2. After the second self-attention layer, consider the last occurrence of any symbol $a$ in the output (say at some position $i > n$, as given in (B.19)): since $n_{a,[0,n]} = n_{a,[n+1,i]}$ by our assumption, decoding this embedding puts most of its mass along $a$ (due to $\widehat{e}'_a$) and its successor (due to $\varepsilon \frac{\sum_{b>a} \alpha_b n_{b,[0,i]} \widehat{e}'_b + \sum_{b \leq a} n_{b,[0,i]} \widehat{e}'_b}{\sum_{b>a} \alpha_b n_{b,[0,i]} + \sum_{b \leq a} n_{b,[0,i]}}$). Again, this "Identity+Successor" behavior shows up in the experiments.

3. Finally, the last MLP layer nullifies the mass on the $a$ token itself, thereby leaving most of the mass on the successor. This aspect also shows up in the experiments.

## C Task Hinting for Other Problems

In this section we discuss the effectiveness of our proposed approach for two problems: that of incrementing a positive integer, i.e., adding 1 to it and the LEGO task that was proposed in Zhang et al. (2022).

### C.1 Increment Task

As we will see, it is quite challenging for transformers to be able to generalize on unseen lengths even for this simple setting of incrementing a number by 1. We again train decoder-only models that produce one token at a time. Similar to the case of sorting, we use the $\perp$ token to denote the end of the input sequence. Each example in the training set is a sequence of the form: $[1, 2, 3, \perp, 4, 2, 1]$, where the output is being produced in *reverse order*, given that is the way in which humans tend to solve this task. The training set contains $1M$ instances of lengths up to 10. Similar to the case of sorting, we skew the distribution towards shorter sequences by sampling $80\%$ of the instances from lengths up to 4. Finally, we ensure that $10\%$ of the samples end with a random sequence of 9s, since these instances are important for the model to learn the notion of a *carry*.

Solving the increment task via a causal decoder-only network presents a different set of challenges than sorting—the instance is no longer permutation-invariant, and as the number of output tokens increases, the model has to attend to a specific position farther to the left in the input sequence. We compare the length-generalization properties of models obtained via standard training versus those obtained via either task hinting or via introducing the tempered softmax operation. For task hinting, we consider the natural hint of making the model output the *carry* sequence along with the output sequence. Hence an instance from the auxiliary task will be structured as

$$[1, 2, 3, \perp, 4, \uparrow, 0, 2, \uparrow, 0, 1, \uparrow, 0],$$

where the $\uparrow$ token represents the fact that the model should output the correct carry value at the next step. We train depth-four transformer models for this task and evaluate their test accuracy on the task of solving the increment problem correctly.

| ↓ Model, → n | 11 | 12 | 13 | 14 | 15 | 16 | 17 | 18 | 19 | 20 |
|---|---|---|---|---|---|---|---|---|---|---|
| Standard | 98.2 | 93.8 | 81.5 | 60.1 | 41 | 23.2 | 10 | 4.1 | 1.4 | 0.3 |
| Hinting | 99.4 | 96.4 | 88.4 | 69.2 | 47.7 | 27.1 | 13.7 | 6 | 2.2 | 0.5 |
| Temp. softmax | 99.8 | 97.5 | 91.4 | 78.3 | 62.1 | 46.4 | 29 | 16 | 8 | 4 |

*Table 3: Test accuracy comparison of various models on the increment task.*

Table 3 compares the performance of the model trained via standard training to (a) the model trained via task hinting, and (b) the model trained using the tempered softmax. We observe that while task hinting helps improve length generalization, the improvements are smaller compared to the improvements for sorting. However, we observe that the model based on tempered softmax helps improve the length generalization to a much greater extent.

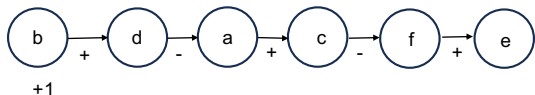

*Figure C.17: An example linear chain for the LEGO task.*

[BOS] a=-d; f=-c; c=+a; d=+b; b=+1; e=+f [EOS]

*Figure C.18: An example input to the model for the chain in Figure C.17. Here [BOS] and [EOS] are special tokens for the beginning and the end of the input.*

## C.2   LEGO TASK

The LEGO task as proposed in the work of Zhang et al. (2022) is a variable resolution task. Given a set of variables, say $\{a, b, c, d, e, f\}$, there is an unknown hidden linear ordering among them. For instance Figure C.17 shows a possible linear ordering. Each variable is assigned a value in $\{-1, +1\}$. The value of the first variable in the chain is provided. The value of any subsequent variable can be obtained by traversing through the chain and flipping the sign of the edge is labeled by "-". Hence for the example shown in Figure C.17 the variable assignments are $\{b = +1, d = +1, a = -1, c = -1, f = +1, e = +1\}$. For a given hidden chain, the input to the model is a sequence of variable assignments describing the input but permuted in a random order. Figure C.18 shows a possible input to the model given that the hidden chain is the one in Figure C.17. Given the input the model is the required to predict the values of all the variables.

We consider the setup described in Zhang et al. (2022) where the authors consider inputs of 12 variables, i.e., chains containing 12 variables. Furthermore, the authors train the model to predict the first 6 variables in the chain, and at inference time evaluate how well does the model generalize to "unseen" lengths, i.e. it's ability to predict variables 7 to 12. We consider whether training the models via task hinting can improve the generalization performance for this task. We consider a natural hint where we additionally provide the model with the value of a random variable (among the first 4 variables in the chain) and then ask it to predict the value of the variable two steps down the chain. Note that this is an easier task then resolving the values of all the variables. Furthermore, we still ensure that during training we do not provide the model with any supervision regarding its performance on variables 7 to 12. Figure C.19 shows an example hinting task for the input described in Figure C.18.

We train decoder only models of depth-12 both with and without task hinting. As expected both the models achieve $100\%$ test accuracy for the first 6 variables. Hence, in Table 4 we report the test accuracy of the models for variables 7 to 12. As we can see, the use of task hinting significantly improves the length generalization for the unseen variables.

| $\downarrow$ Model, $\rightarrow$ n | 7 | 8 | 9 | 10 | 11 | 12 |
|---|---|---|---|---|---|---|
| Standard | 99.9 | 99.4 | 94.8 | 84.5 | 67.6 | 41.3 |
| Hinting | 99.9 | 99.9 | 98.4 | 91.9 | 79.3 | 56.5 |

*Table 4: Test accuracy comparison of the models on the LEGO task.*

## D   EXPERIMENTAL SETTINGS AND HYPERPARAMETERS

**Sorting Dataset construction.** For the sorting problem, we construct the data set for the main task by sampling a length in $\{2, 3, \ldots, 20\}$ from a skewed distribution. For a given length, we sample an input sequence by independently drawing random numbers in $\{1, 100\}$ uniformly at random with replacement, for each input position. The skewed length distribution places $80\%$ of the probability mass equally on lengths in $\{2, 3, 4, 5\}$ and the remaining $20\%$ uniformly on lengths in $\{6, 7, \ldots, 20\}$.

We also train models on variants of the above dataset that contain non-trivial repetitions. These datasets are created by first picking a length $\ell$ from the same skewed distribution. With probability

[BOS] a=-d; f=-c; c=+a; d=+b; b=+1; e=+f [EOS] [BOH] a=-1; f=? [EOH]

*Figure C.19: An example hinting task to the model for the input in Figure C.18. Here [BOH] and [EOH] are special tokens for the beginning and the end of the hint.*

0.9, we follow the same procedure as above for creating the input sequence. With the remaining probability $0.1$, we pick $\frac{\ell}{2}$ numbers uniformly at random from $\{1, 2, \ldots, 100\}$ (without replacement) and create the input sequence by independently drawing numbers from this set (uniformly, with repetitions) for each position in the input sequence.

The training dataset for the *successor hint* is created in the same manner as above: having picked the input sequence, we pick a position uniformly at random and use the corresponding number to create the successor hint. For creating the *count hint* dataset we again pick the length of the sequence from the skewed distribution. Then we pick two numbers $a, b$ at random (without replacement) from $\{1, 2, \ldots, 100\}$. Finally, with probability $0.5$ we repeat them the same number of times, and with the remaining probability either $a$ or $b$ is chosen (equally) to be the under-represented number. The amount of under-representation is chosen uniformly among the valid choices, but restricted to be at most $5$. Similarly, for the *fill task*, we choose the length $\ell$ from the skewed distribution and choose to repeat the element some number of times uniformly chosen from $\{1, 2, \ldots, \lfloor \ell/2 \rfloor\}$. To construct the test set for each sequence length, we sample $100k$ examples uniformly at random with replacement.

All our models for the sorting network use the architecture and parameters detailed in Table 5. We do not use position embeddings in our architectures, as causal decoder-only models are not permutation invariant. In all our experiments with using fixed and learned positional embeddings, we observed comparable or even worse performance compared to models using no positional embedding.

| Parameter | Value |
|---|---|
| Embedding size $d$ | 1024 |
| Vocabulary size $q$ | 103 |
| Position embedding type | None |
| # Attention heads $h$ | 16 |
| MLP inner dimensionality $d'$ | 2048 |
| Sequence length | 512 |
| Base learning rate | 1e-5 |
| Optimizer | Adam |
| LR warmup | Linear for 10 epochs |
| LR decay schedule | Cosine, one cycle with default parameters |
| Dropout | None |
| Activation | GELU |

*Table 5: Hyperparameters for the sorting task.*

**Increment Dataset Construction.** For the problem of incrementing a positive integer, we construct the data set for the main task by sampling a length in $\{2, 3, \ldots, 10\}$ from a skewed distribution. The distribution places $80\%$ mass equally on lengths $\{2, 3, 4\}$ and the remaining $20\%$ equally on lengths $\{5, \ldots, 10\}$. For each length $\ell$, with probability $0.9$ we sample input position $0$ (the most significant digit) uniformly at random from $\{1, \ldots, 9\}$ and the remaining positions uniformly at random (with replacement) from $\{0, \ldots, 9\}$. With the remaining probability of $0.1$, we randomly replace the last $k$ positions with the digit $9$, where $k$ is chosen uniformly from $\{1, 2, \ldots, \ell\}$. Our test dataset for each length consists of $100k$ numbers chosen uniformly at random. For the increment task we use the parameters detailed in Table 7 and always train depth-4 models.

**LEGO Dataset Construction.** We follow the same setup as described in Zhang et al. (2022). To generate a given input of 12 variables, we first randomly select 12 variables from $\{a, \ldots, z\}$ uniformly without replacement and then randomly permute them to get the hidden chain. We label each edge of the chain randomly to be $\{-1, +1\}$ and we assign a random value in $\{-1, +1\}$ to the variable at the beginning of the chain. Then we form the input example by considering a random permutation of the 12 different variable assignments. Our total training set is of size $1M$ and our test set is of size $100,000$. Note, that the work of Zhang et al. (2022) considered training sets of size

| Parameter | Value |
|---|---|
| Embedding size $d$ | 1024 |
| Vocabulary size $q$ | 14 |
| Position embedding type | None |
| # Attention heads $h$ | 16 |
| MLP inner dimensionality $d'$ | 2048 |
| Sequence length | 512 |
| Base learning rate | 1e-5 |
| Optimizer | Adam |
| LR warmup | Linear for 10 epochs |
| LR decay schedule | Cosine, one cycle with default parameters |
| Dropout | None |
| Activation | GELU |

*Table 6: Hyperparameters for the increment task.*

$120,000$. However we found that even the standard models have high variance in the performance at this scale of dataset size. Furthermore, as is done for the sorting task and the increment task we train decoder-only models as opposed to the encoder models trained in Zhang et al. (2022).

| Parameter | Value |
|---|---|
| Embedding size $d$ | 1024 |
| Vocabulary size $q$ | 64 |
| Position embedding type | None |
| # Attention heads $h$ | 16 |
| MLP inner dimensionality $d'$ | 4096 |
| Sequence length | 128 |
| Base learning rate | 1e-5 |
| Optimizer | Adam |
| LR warmup | Linear for 10 epochs |
| LR decay schedule | Cosine, one cycle with default parameters |
| Dropout | None |
| Activation | GELU |

*Table 7: Hyperparameters for the LEGO task.*

# E OTHER BASELINES

In this section we compare the framework of task hinting with other natural baselines for improving length generalization on the sorting task. The first method we compare against is the idea of *curriculum learning* that has been explored in recent works (Jelassi et al., 2023; Abbe et al., 2023). However, curriculum learning typically requires one to successively introduce instances from higher and higher lengths. Since our goal is to study whether we can truly generalize from training on only short sequences (of length up to 20), we implement curriculum learning by dividing our training set for the sorting task (as described in Section D) into sets $S_4, S_8, S_{12}, S_{16}, S_{20}$ where $S_i$ contains sequences of length up to $i$. We divide the total number of training epochs into four stages and in each stage progressively introduce higher and higher lengths.

| ↓ Model, → n | 50 | 100 |
|---|---|---|
| Standard | 98 | 0 |
| Curriculum | 98.4 | 1.2 |
| Scratchpad | 98.2 | 0 |
| Hinting | 99.8 | 92.6 |

*Table 8: Test accuracy comparison of the baselines and the hinting method on the sorting task.*

Input:  5  3  1  4  2  101  [BOS] [N] 1  [N] 2 [N] 3  [N] 4 [N] 5 [EOS]

*Figure E.20: An example input for scratchpad training. Here [BOS] and [EOS] are special tokens for the beginning and the end of the scratchpad. The [N] token is a special token for producing the next element given the input and the current scratchpad.*

The second methodology we compare against is the idea of *scratchpad training* that was utilized in the work of Anil et al. (2022) for the parity task with limited success. For the sorting task, since the decoder only model is learning to solve the task by producing one element at a time, we consider the natural scratchpad which defines all the successor elements that have been output by the model so far. See Figure E.20 for an example.

We follow the same training procedure as described in Section 3. Table 8 shows the performance of the three methods for lengths higher than 20. As we can see, both curriculum training and scratchpad training do no better than the standard training for length generalization and only the task hinting based model achieves non-trivial performance for length 100.

