# OpenReview forum: "Improving length generalization in transformers via task hinting"
_ICLR.cc/2024/Conference — Submitted to ICLR 2024_

### Official Review · Reviewer_bvbc · 2023-10-31

**Soundness:** 3 good
**Presentation:** 3 good
**Contribution:** 2 fair
**Rating:** 5
**Confidence:** 3

**Summary:**

This paper proposes to improve the length generalization properties of the transformer by simultaneously training the model to solve a simpler but related auxiliary task named task hinting. This paper studies the classical sorting problem and demonstrates that models trained via task hinting significantly improve length generalization.

The authors also show that the effectiveness of different tasks hinting at improving length generalization differs dramatically. They further use probing and visualization-based techniques to understand the internal mechanisms, and propose a theoretical construction consistent with the observed learning behavior of the model.

**Strengths:**

The paper proposes a new method for improving length generalization and demonstrates it experimentally.

**Weaknesses:**

1. It is not a surprising phenomenon that simultaneously training the model with some related task can help to improve the generalization capability of transformers.
2. The proposed method to solve length generalization is not generic. For any specific task, "task hinting" requires researchers to search for a good task that could improve length generalization. Searching for such a task may be complicated and highly task-dependent.
3. The authors only studied the sorting task (and another task hidden in the appendix). It is unclear if "task hinting" could be easily developed as a generic method for solving length generalization.

**Questions:**

Could the authors respond to my comments above?

---

> ### Author Response · Authors · 2023-11-16
> **Response to comments**
>
> Thanks for the review. We address your points below.
>
> >> It is not a surprising phenomenon that simultaneously training the model with some related task can help to improve the generalization capability of transformers.
>
> We argue that the situation is more nuanced. While it does seem natural that related tasks should help, we found it surprising that some tasks that a-priori seem quite natural do not help length generalization, whereas others do! In particular, for the sorting problem the successor task helped the most, whereas the counting and fill tasks (which are also natural subtasks) did not help almost at all. Our investigations in Sections 3 and 4 were directly motivated by this surprising difference in performance: we aimed to provide a more detailed analysis of why some particular tasks were so useful.
>
> Another thing that surprised us was the extent to which task hinting helped. For instance, the  accuracy jumped from ~0% to 92% when considering length 100 sequences. In fact, even for sequences with length 500, our models gave outputs with very small edit distances, having been trained on sequences of length only up to 20! These are the first instances we know of where such extreme length generalization has been observed.
>
> >> The proposed method to solve length generalization is not generic. For any specific task, "task hinting" requires researchers to search for a good task that could improve length generalization. Searching for such a task may be complicated and highly task-dependent.
>
> We agree with you that in general, the problem of finding appropriate tasks may itself be quite challenging. (E.g., we discuss this in the limitations section in Appendix A.) Our hope in doing a comprehensive study for the sorting problem (and particularly for doing the interpretability analysis) is to better understand when and how task-hinting is helpful. Based on our analysis in Section 4 we posit that tasks that align well with the internal learning biases of the model are the ones that help the most. Understanding automated/algorithmic ways to find such tasks for an arbitrary setting is a fascinating direction for future research.
>
> >> The authors only studied the sorting task (and another task hidden in the appendix). It is unclear if "task hinting" could be easily developed as a generic method for solving length generalization.
>
> To demonstrate the effectiveness of our proposed task-hinting framework beyond sorting, we have added results for the LEGO task, which has been studied in recent works in the context of generalization. These results are given in Appendix C.2, and again demonstrate the benefits of task hinting. Briefly, the LEGO task consists of a variable resolution problem over 12 variables where during training the model is only provided supervision for the first $n=6$ variables. We then measure the accuracy of the model in predicting the correct values for variables 7 to 12. For easier readability we include the results in the table below as well. Please see Appendix C.2 for more details.
>
> | n->    | 7 |  8 |  9 |  10 |  11 |  12 |
> | -------- | ------- |  ------- |  ------- |  ------- |  ------- |  ------- |
> | Standard  | 99.9    |  99.4    |  94.8    |  84.5    |  67.6    |  41.3   |
> | Hinting |    99.9     |  99.9    |  98.4    |  91.9    |  79.3    |  56.5    |

---

### Official Review · Reviewer_9TMa · 2023-11-03

**Soundness:** 3 good
**Presentation:** 3 good
**Contribution:** 2 fair
**Rating:** 3
**Confidence:** 5

**Summary:**

This paper tackles transformers' length generalization issue by introducing task hinting, a method that trains the model on an auxiliary task alongside the main task. Using the sorting problem as a test case, the authors demonstrate improved length generalization. They also explore the internal workings of the model. They have constructed a sorting network, but it's not directly related to the task hinting idea.

**Strengths:**

The paper is original in its approach to task hinting, a relatively unexplored idea. The idea seems promising from the current results, but only applied to a single task. The paper is generally clear, although there are areas where further clarification would be beneficial.

**Weaknesses:**

- The paper's main weakness is its limited application to a single task, which weakens the validity of the proposed idea.
- It's also unclear if the same performance gain could be achieved via curriculum learning.
- Also, the comparison between this approach and scratchpad training is not made. What if we provide auxiliary hints as a part of the scratchpad?
- The theoretical results are *not* related to the key idea of task hinting.

**Questions:**

- It's unclear if the same performance gain could be achieved via curriculum learning. In some sense, the auxiliary task can be viewed as a partial sorting, i.e., so it's a simpler task than the original target task.

- Could you clarify the role of the mask values in Figure 3.1? Why are there mask values of 0 after some 1's? Not sure why you need to pad when training decoder models.

- Could you provide more details on the fill hint task? Its current explanation is unclear.

- Does this idea work for tasks other than sorting? Could you apply the same to many other tasks?

- Can you use the factor analysis (e.g., those introduced in the recent paper [1]) for the interpretation part? Especially see the induction head analysis of this aforementioned paper.

[1] https://arxiv.org/abs/2310.04861

---

> ### Author Response · Authors · 2023-11-16
> **Response to comments**
>
> Thanks for the review. We address your points below.
>
> >> It's unclear if the same performance gain could be achieved via curriculum learning. In some sense, the auxiliary task can be viewed as a partial sorting, i.e., so it's a simpler task than the original target task.
>
>
> Thanks for your suggestions for other methods such as curriculum learning and scratchpad training. Typically in curriculum learning, one has to progressively include sequences of higher and higher lengths. Our goal instead is to understand whether we can generalize from training on short sequences alone (length up to 20). Nonetheless, we have included curriculum learning as a baseline in the updated draft.
>
> Regarding scratchpad training, the work of Anil et al.’22 observed that fine-tuning on scratchpad data only helps length generalization to a limited extent. Is there a particular scratchpad that you had in mind? We had trouble coming up with meaningful scratchpads for the sorting problem that were length-efficient. (E.g., we considered adding all the steps of some sorting algorithm like merge-sort or insertion-sort as a scratchpad, but these blow up the input length considerably, making it super inefficient for both training and inference. Our approach of training via subtasks does not suffer from this issue.) In Appendix E we have added a comparison to a length-efficient scratchpad—however, this scratchpad is not particularly different from the natural input/output representation. We would be interested in hearing what scratchpads you think might be useful to compare to.
>
> To summarize, we followed your suggestion and added comparisons to the above two methods. The results are in Appendix E of the updated draft. These experiments show that task hinting is the only method in this collection which achieves non-trivial performance for sequence lengths beyond 50. For easier readability the results are also included in the table below.
>
> | n->    | 50 | 100 |
> | -------- | ------- |------- |
> | Standard  | 98    |0    |
> | Curriculum  | 98.4    |1.2    |
> | Scratchpad | 98.2     |0     |
> | Hinting   | 99.8    |92.6    |
>
>
> >> Could you clarify the role of the mask values in Figure 3.1? Why are there mask values of 0 after some 1's? Not sure why you need to pad when training decoder models.
>
> A mask value of 1 means that we penalize the predictions made at the corresponding position during the loss function computation. Since we use padding, we use a mask value of 0 to avoid adding terms corresponding to these padding positions into our loss function. (The use of padding is simply for ease of training: the padding makes each training example be the same length.) We will clarify this further in the final version.
>
> >> Could you provide more details on the fill hint task? Its current explanation is unclear.
>
> Thanks for the question. The motivation for the fill hint task is the following: to sort data with repeated elements, a sorting algorithm must be able to keep track of the number of copies of an element that have been output already, and the number still to be output. Instances of the fill task consist of a number, say 17, repeated some number k (say k=5) times, followed by a prefix consisting of the same number 17 repeated r < k times. The goal of the task is to teach the model that k-r copies of the number 17 still need to be output. We will make this discussion clearer in the final version.
>
> >> Does this idea work for tasks other than sorting? Could you apply the same to many other tasks?
>
> To demonstrate the effectiveness of our proposed task-hinting framework beyond sorting, we have added results for the LEGO task, which has been studied in recent works in the context of generalization. These results are given in Appendix C.2, and again demonstrate the benefits of task hinting. Briefly, the LEGO task consists of a variable resolution problem over 12 variables where during training the model is only provided supervision for the first $n=6$ variables. We then measure the accuracy of the model in predicting the correct values for variables 7 to 12. For easier readability we include the results in the table below as well. Please see Appendix C.2 for more details.
>
> | n->    | 7 |  8 |  9 |  10 |  11 |  12 |
> | -------- | ------- |  ------- |  ------- |  ------- |  ------- |  ------- |
> | Standard  | 99.9    |  99.4    |  94.8    |  84.5    |  67.6    |  41.3   |
> | Hinting |    99.9     |  99.9    |  98.4    |  91.9    |  79.3    |  56.5    |
>
> >> Can you use the factor analysis (e.g., those introduced in the recent paper [1]) for the interpretation part? Especially see the induction head analysis of this aforementioned paper.
>
> Thanks for the reference! This work seems to be subsequent to the submission of our paper. We will definitely look into it and investigate whether this analysis can be applied to our setting to further demystify the mechanisms of the learned models.

---

### Official Review · Reviewer_mVzC · 2023-11-04

**Soundness:** 2 fair
**Presentation:** 1 poor
**Contribution:** 2 fair
**Rating:** 3
**Confidence:** 4

**Summary:**

The paper proposes a method called "task hinting" to improve how well transformer models generalize to longer sequences than they were trained on. This involves training the transformer on both the main task and a related simpler task simultaneously. The authors found that this method significantly improves the model's performance on longer sequences. The effectiveness of different auxiliary tasks varies, with some aiding generalization more than others. Additionally, a small number of length-dependent parameters introduced into the model can further enhance performance.

**Strengths:**

1: The finding of this paper is interesting. The visualized internal mechanism is surprising if there is no cherry-picking.
2: The proposed task hinting method looks good considering its effectiveness on the sorting tasks.

**Weaknesses:**

1: Multi-task is not novel.  How / why it can help length generalization is still unclear in this paper.
2: The writing is poor. This paper is not organized well. For example, at the beginning, authors try to categorize length generalization into two categories. However, the taxonomy does not make sense and it cannot fit the task-hinting well.
3: The results of tempered softmax are not surprising. With longer sequences, self-attention requires scaling is well-known. (The hugging face transformer library has even involved this feature). To be more specific, the scaling factor should be "log_n scale'.
4: This paper is not completed within the required number of pages by ICLR.
5: Some typo and grammatical errors.

**Questions:**

Please check the weakness part.

---

> ### Author Response · Authors · 2023-11-16
> **Response to comments**
>
> Thanks for the review. We address your points below.
>
> >> Multi-task is not novel. How / why it can help length generalization is still unclear in this paper
>
> We do not claim that we are proposing multitask learning as a new paradigm. Our main contribution is the idea of using auxiliary task-hinting to improve length generalization. In order to incorporate auxiliary tasks into the training setup, we use multitask learning via fairly standard setups. This is already discussed in Appendix A.
>
> As to the question of how/why it helps length-generalization: this is exactly the goal of Section 4. That section is devoted to understanding why task hinting helps, and why certain tasks are better than others. We use our interpretability analysis to posit a concrete conjecture, which says that tasks which align well with the internal learning biases of the model are the ones that help the most for length generalization. We show some corroborating evidence for the case of the sorting problem.
>
> >> This paper is not organized well. For example, at the beginning, authors try to categorize length generalization into two categories. However, the taxonomy does not make sense and it cannot fit the task-hinting well
>
> Perhaps there is some confusion here. Our goal is to characterize existing approaches to length generalization into two categories: (a) approaches that finetune an existing pre-trained LLM and (b) those that consider specific problems and train models for these specific problems from scratch. In this sense, our work belongs to the second category: it studies specific problems and trains models from scratch. We can reword this in the final version to further clarify this.
>
> >> The results of tempered softmax are not surprising. With longer sequences, self-attention requires scaling is well-known. (The hugging face transformer library has even involved this feature). To be more specific, the scaling factor should be "log_n scale'.
>
> Tempered softmax is a natural operation that comes out of our theoretical analysis: we then make this a trainable parameter for each layer in our experiments. However, this investigation of tempered softmax is secondary to the main results of the paper. If an existing library indeed incorporates it, we view this as a positive for our theoretical work, which provides further evidence for it being a good idea. On that note, we searched the HuggingFace library to find uses of “log_n scale” which you mentioned, but we were unsuccessful. If you could please point us to the right online references, we would be happy to connect our work to it, and also provide further clarifications on any similarities and differences.
>
> >> This paper is not completed within the required number of pages by ICLR.
>
>
> We respectfully disagree. The main paper is within the first 9 pages followed by the citations and the appendix. The call for papers clearly permits the use of an arbitrary number of pages for the supplementary material in the appendices.

---

### Official Review · Reviewer_rpHo · 2023-11-08

**Soundness:** 4 excellent
**Presentation:** 4 excellent
**Contribution:** 3 good
**Rating:** 6
**Confidence:** 4

**Summary:**

The paper considers the role of introducing auxiliary tasks during training for the problem of length generalization. In particular, they focus on the task of sorting. When trained on sequences of length up to 20, the model gets less than 1% accuracy on sequences of length 100 (and scaling training dataset size or model size doesn't help much). However, when the training dataset is augmented with an auxiliary and simpler task of predicting the successor of a given number in the sorted list, the accuracy at length 100 jumps significantly. Not surprisingly, the efficacy of this method varies between different auxiliary tasks.

Further, the paper also suggests a theoretical construction for how a 2-layer Transformer might be able to do this task. The theoretical construction uses a length-dependent parameter used for the softmax operation in attention layers. The introduction of this length-dependent parameter is empirically shown to help in length generalization (even in the absence of auxiliary tasks).

**Strengths:**

1. I enjoyed reading the paper. It is very well written.

2. The idea of using auxiliary data to improve length generalization seems very natural and this paper takes the first step in exploring this in a simple setting. It can potentially lead to more interesting work on how to select the auxiliary data.

3. The idea of using a length-dependent parameter in the softmax operation also seems interesting and deserves more attention.

**Weaknesses:**

One major weakness of the paper is that its scope is a bit limited. While it shows the efficacy of using auxiliary data for length generalization, its results are limited largely to the problem of sorting. While I don't doubt that auxiliary data can help in length generalization for other tasks as well, it is unclear how to go about finding the right auxiliary data given a task. Nevertheless, the paper takes a step in this direction and can lead to interesting future work.

**Questions:**

In this paper, no positional embeddings seem to have been used, does the efficacy of the proposed auxiliary task change when using positional embeddings?

---

> ### Author Response · Authors · 2023-11-16
> **Response to comments**
>
> Thank you for your comments and for your appreciation of the work. To demonstrate the effectiveness of our proposed task-hinting framework beyond sorting, we have added results for the LEGO task, which has been studied in recent works in the context of generalization. These results are given in Appendix C.2, and again demonstrate the benefits of task hinting. Briefly, the LEGO task consists of a variable resolution problem over 12 variables where during training the model is only provided supervision for the first $n=6$ variables. We then measure the accuracy of the model in predicting the correct values for variables 7 to 12. For easier readability we include the results in the table below as well. Please see Appendix C.2 for more details.
>
> | n->    | 7 |  8 |  9 |  10 |  11 |  12 |
> | -------- | ------- |  ------- |  ------- |  ------- |  ------- |  ------- |
> | Standard  | 99.9    |  99.4    |  94.8    |  84.5    |  67.6    |  41.3   |
> | Hinting |    99.9     |  99.9    |  98.4    |  91.9    |  79.3    |  56.5    |
>
> We agree with you that in general, the problem of finding appropriate tasks may itself be quite challenging. (E.g., we discuss this in the limitations section in Appendix A.) Our hope in doing a comprehensive study for the sorting problem (and particularly for doing the interpretability analysis) is to better understand when and how task-hinting is helpful. Based on our analysis in Section 4 we posit that tasks that align well with the internal learning biases of the model are the ones that help the most. Understanding automated/algorithmic ways to find such tasks for an arbitrary setting is a fascinating direction for future research.
>
> Regarding positional embeddings, we did not find any changes in the efficacy of the tasks when positional information is present vs. when it is not. Indeed, for the sorting problem, the successor task continued to be the best among the three types of tasks considered.

---

> > ### Comment · Reviewer_rpHo · 2023-11-21
> >
> > I thank the authors for answering my questions and running the additional experiment. I will take this into account while engaging with other reviewers and deciding the final score for the paper.

---

### Author Response · Authors · 2023-11-20
**Looking forward to the discussion following our response!**

Dear reviewers,

Since the discussion period ends in 2 days, it would be great to hear if there are other concerns that we can discuss. We hope the responses have addressed the concerns that the reviewers raised in their original reviews. We have also updated the paper draft and included new experiments based on the reviewers' comments. Please let us know. Looking forward to hearing from you.

---

### Meta-Review · Area_Chair_tYfw · 2023-12-13

**Metareview:**

This paper proposes task hinting (simultaneously training on a related and simpler task) to improve the length generalization of transformers. The paper demonstrates this on the classical sorting problem, for which task hinting leads to significant improvement in length generalization. The paper further does interpretability analyses and proposes a theoretical construction consistent with the observed learning behaviors of the model.

The reviewers found the results interesting and agreed that task hinting looks like a natural and promising method to improve length generalization. A couple of major concerns were pointed out by the reviewers. First, the paper is limited to a single task (sorting) and it's unclear whether task hinting works for other tasks. To address this, the authors provided another experiment on the LEGO task. Second, the paper doesn't discuss how to find a good auxiliary task, which limits the applicability of the proposed task hinting method.

**Justification For Why Not Higher Score:**

While task hinting is an interesting idea and the presented result is promising, there isn't a generic way or principle to find a good auxiliary task, which limits its applicability.

**Justification For Why Not Lower Score:**

N/A

---

### Decision · Program_Chairs · 2024-01-16

Reject